# A Thai Traditional Triple-Fruit Formulation “Phikud Tri-Phon” May Provide Fat Loss and Nutritional Benefits

**DOI:** 10.3390/foods11193067

**Published:** 2022-10-02

**Authors:** Ngamrayu Ngamdokmai, Kornkanok Ingkaninan, C. Norman Scholfield, Kamonlak Insumrong, Nitra Neungchamnong, Genet Minale, Sakan Warinhomhoun

**Affiliations:** 1School of Medicine, Walailak University, Nakhon Si Thammarat 80160, Thailand; 2Center of Excellence in Marijuana, Hemp, and Kratom, Walailak University, Nakhon Si Thammarat 80160, Thailand; 3Centre of Excellence in Cannabis Research, Center of Excellence for Innovation in Chemistry, Department of Pharmaceutical Chemistry and Pharmacognosy, Faculty of Pharmaceutical Sciences, Naresuan University, Phitsanulok 65000, Thailand; 4Akkhraratchakumari Veterinary College, Walailak University, Nakhon Si Thammarat 80160, Thailand; 5Department of Chemistry, Faculty of Science, Naresuan University, Phitsanulok 65000, Thailand; 6Science Laboratory Centre, Faculty of Science, Naresuan University, Phitsanulok 65000, Thailand; 7Department of Pharmacy, College of Health Sciences, Ambo University, Ambo P.O. Box 19, Ethiopia

**Keywords:** Tri-Phon, Thai herbal medicine, antioxidant, anti-adipogenesis, obesity, uncoupled fat metabolism

## Abstract

Obesity and overweight have serious health outcomes. “Phikud Tri-Phon” (PTP) is a traditional Thai medicine comprising three dried fruits from *Aegle marmelos* L., *Morinda citrifolia* L., and *Coriandrum sativum* L. Whether this medicine impacts on metabolic disease is unclear. This study aimed to investigate the phenolic and flavonoid contents of PTP and each of its herbal components, and further assess their antioxidant and anti-adipogenetic activities. Oil-red O staining was measured for lipid accumulation in 3T3-L1 adipocytes. The chemical profiles of PTP and each herbal extract were determined by LC-ESI-QTOF-MS/MS. Our results show that the total phenolic and flavonoid contents of PTP water extract were 22.35–108.42 mg of gallic acid equivalents and PTP ethanolic extract was 1.19–0.93 mg of quercetin equivalents and the DPPH scavenging capacity assay of PTP ethanolic extract (1 mg/mL) was 92.45 ± 6.58 (Trolox equivalent)/g. The PTP extracts and individual herbs had inhibitory adipogenesis activity, which reduced lipid accumulation by approximately 31% in PTP water extract and 22% in PTP ethanolic extract compared with control cells. These results provided insights into the traditional preparation method of using boiling water as a vehicle for PTP. In conclusion, PTP has antioxidant and anti-adipogenesis potential, indicating it is a promising ingredient in functional food and herbal health products.

## 1. Introduction

Obesity results from an imbalance between energy intake and expenditure and has become a risk factor for a wide spectrum of serious health issues [1,2]. The major manifestation of obesity is excess adiposity, leading to cancers, diabetes mellitus, cardiovascular disease, chronic kidney disease, metabolic syndrome, and a myriad of other conditions [3]. Diets are dominated by long-chain saturated fatty acids, high glycemic starches and monosaccharides, and food processing that have high energy content but poor nutrition. On the other side of the energy balance, reduced physical activity fails to rectify the positive energy imbalance [4,5], and energy is stored as lipids in adipocytes accommodated through either adipocyte hyperplasia or hypertrophy [6]. Obesity is characterized at the cellular level by an increase in adipogenesis and is the most important energy store for humans [7]. Nevertheless, the storage capacity of career adipocytes is limited and their further loading with triglycerides causes adipose tissue to become inflammatory, plasma triglyceride concentrations to increase and the stored fat can no longer be mobilized [8]. Accordingly, excess potentially toxic fat is sequestered by ectopic non-adipocytes that interfere with their normal dedicated functions [9,10]. Obesity prevention has attracted much research effort, especially through stimulating the decomposition of accumulated fat. Lipid droplets in adipocytes, therefore, are considered an essential factor in the regulation of lipid metabolism and accumulation, and the regulation of triglyceride synthesis and lipolysis [11,12]. To overcome these problems, several phytochemicals such as quercetin, epigallocatechin-3-gallate, resveratrol, caffeic acid, and gallic acid have been shown to mobilize triglycerides with low toxicity [9,13].

Thai traditional medicines (TTM) include many indigenous medical practices and formulations long used in Thailand and which continue to be practiced today. Thai herbal medicine (THM) falls into two groups: remedies from a single plant and herbal remedies of complex formulae. The main principles of THM include treating the cause of the illness and the body elements [14]. TTM includes many medicinal formulae that are prepared from various combinations of herbal ingredients. These combinations were believed to produce maximal therapeutic efficacy with fewer associated side effects or toxicity [15].

Phikud Tri-Phon (PTP) is traditional medicine containing fruits/seeds of *Aegle marmelos* L. Correa or Bael tree (Rutaceae), *Morinda citrifolia* L. or Noni (Rubiaceae), and *Coriandrum sativum* L. or Coriander (Apiaceae) in equal amounts. Traditionally, it is used as anti-emetics or purgatives. *A. marmelos* fruit extract promoted lipolysis in cultured adipocytes [16] and reduced blood glucose in diabetic or high fat-fed rats [17]. Two compounds, marmin and fagarine, are contained in leaves, seeds, and fruits of *A. marmelos*. Methanolic extract of bark is particularly rich in both compounds and are responsible for male anti-fertility actions. Several studies using in vitro models reported reduced sperm viability, motility, density, and acrosomal integrity [18,19,20,21]. *M. citrifolia* extract stimulates glucose uptake into 3T3-L1 adipocytes while also reducing carbohydrate digestion and glucose absorption through α-amylase and α-glucosidase inhibition [22,23,24], suggesting that it may reduce starch digestion and glucose absorption. *M. citrifolia* wine slowed weight gain [25,26]. *C. sativum* extract reduced triglyceride formation in 3T3-L1 cells and ameliorated insulin resistance and adipocyte hypertrophy [27,28]. Thus, a combination of these ingredients as formulated into PTP is likely to curtail lipogenesis. However, since PTP contains multiple ingredients, we query whether all the three components contribute to overall therapeutic and nutritional actions.

This study aimed to (i) assess in vitro the therapeutic anti-adipogenic potential and the general nutritive functions fulfilled by the antioxidants, phenolics, and flavonoids of PTP and its three separate ingredients, and (ii) to determine whether all three ingredients are needed for PTP action or whether the formulation could be simplified. These functions will be assessed by standardized in vitro methods, cultures of 3T3-L1 adipocytes and phytochemical analyses by LC-ESI-QTOF-MS/MS.

## 2. Materials and Methods

### 2.1. Chemicals and Reagents

Gallic acid (GAE), quercetin (QE), 1,1-diphenyl-2-picrylhydrazyl (DPPH), 6-hydroxy-2,5,7,8-tetramethylchroman-2-carboxylic acid (Trolox), L-ascorbate, Folin-Ciocalteu’s reagent, sodium carbonate, sodium acetate, and aluminum chloride were purchased from Sigma-Aldrich (St Louis, MO, USA). Quercetin was purchased from TCI (Tokyo, Japan). Caffeine was purchased from Sigma-Aldrich (Buchs, Switzerland). Acetonitrile LC-MS grade was purchased from Labscan, (Bangkok, Thailand). Ultrapure water was prepared by using a millipore MilliQ Integral water purification system (Millipore, Bedford, MA, USA). Formic acid (analytical grade) was obtained from Merck (Darmstadt, Germany). Analytical grade of 95% ethanol was purchased from CHEMIPAN (Bangkok, Thailand).

3-(4,5-Dimethylthiazol-2-yl)-2,5-diphenyl-tetrazolium bromide (MTT), Oil Red O reagent and human recombinant insulin were purchased from Sigma-Aldrich (St. Louis, MO, USA). Dexamethasone (DEX) and isobutylmethylxanthine (IBMX) were purchased from Merck (Kenilworth, NJ, USA). The other chemicals were analytical grade. Bovine calf serum (BCS), fetal bovine serum (FBS), high-glucose Dulbecco’s modified Eagle’s medium (DMEM), phosphate-buffered saline (PBS), trypsin-EDTA and antibiotics (P/S) were purchased from GIBCO (Grand Island, NY, USA).

### 2.2. Plant Materials

Dried ripe fruits of *A. marmelos*, *M. citrifolia*, and mature seeds of *C. sativum* were purchased in August 2021 from a traditional market (P’Aun) at Wat Yai, Phitsanulok, Thailand. They were authenticated by comparing them with voucher lots available in the Biological Sciences Herbarium at Naresuan University, Phitsanulok or comparing them with botanical illustrations.

### 2.3. Extraction of Phikud Tri-Phon (PTP) and the 3 Herbal Products

PTP was formulated by combining 50 mg each of *A. marmelos*, *M. citrifolia*, and *C. sativum* and grinding the mixture into a powder. Crude aqueous and crude ethanolic extracts were produced. For aqueous extracts, the PTP powder and single herbs were extracted with three changes of distilled water boiled for 20 min each. The aqueous extracts were lyophilized to provide 4 dried extracts.

Ethanolic extracts were prepared from the PTP powder and single herbs using 95% EtOH at room temperature for 3 days, and three changes of fresh solvent. Filtrates for each herb were pooled and evaporated under reduced pressure to give 4 ethanolic extracts. All 8 extracts were stored at 20 °C until further use, described as ‘samples’.

### 2.4. Determination of Total Phenolic Contents

Total phenolic content was measured using Folin-Ciocalteu’s method, with some modifications [29,30,31]. Briefly, the 25 µL of extract samples (1 mg/mL) were mixed with 25 µL of Folin-Ciocalteu’s reagent (diluted 1:3 water) and 25 µL of 10% sodium carbonate solution. The mixture was incubated at room temperature in darkness for 1 h. The absorbance was measured at 765 nm using a microplate reader (Hybrid Multi-Mode detection Synergy H1, model H1MF, Bio-TeK Instruments, Santa Clara, CA, USA). All samples were determined in triplicate. Standards contained 1.56–200 µg/mL gallic acid and results were expressed as milligram gallic acid (GAE) equivalent (mg GAE/g extract).

### 2.5. Determination of Total Flavonoid Contents

Total flavonoid contents were determined by an aluminum chloride colorimetric assay [29,32]. Briefly, 50 µL samples (4 mg/mL) were mixed with 10 µL 10% AlCl_3_, 1 M sodium acetate and 150 µL of 95% ethanol. Mixtures were further incubated in the dark at room temperature for 40 min. The absorbance was measured at 415 nm using a microplate reader. Standards contained 1.6–50 µg/mL quercetin, and results were expressed as milligram quercetin (QE) equivalents (mg QE/g extract).

### 2.6. Free Radical Scavenging Using DPPH

The antioxidant activity of samples was assessed by the DPPH assay, with some modifications [32]. 100 µL of extract at 3.91–2000 µg/mL in ethanol was added to 100 µL of DPPH (500 µM in ethanol) in each well of a 96-well plate. The mixture was incubated for 30 min at room temperature in the dark. The absorbance was measured at 515 nm using a microplate reader. L-Ascorbate and Trolox were used as positive controls, and radical scavenging (%) was calculated using the equation:$$\mathrm{Radical}\;\mathrm{scavenging}\;\left(\%\right)=\frac{\mathrm{Absorbance}\;\mathrm{of}\;\mathrm{control}\;–\;\mathrm{Absorbance}\;\mathrm{of}\;\mathrm{test}\;\mathrm{sample}}{\mathrm{Absorbance}\;\mathrm{of}\;\mathrm{control}}\times 100$$

The IC_50_ of each extract was determined from the graph plotted as the percentage of inhibition against concentration of extract (µg/mL).

### 2.7. Cell Culture and Reagents

Mouse embryonic 3T3-L1 pre-adipocytes were obtained from ATCC (Manassas, VA, USA). The 3T3-L1 cells were cultured in a completed media, comprised of Dulbecco’s modified Eagle’s medium (DMEM) supplemented with 10% fetal bovine serum (FBS), 3.7 g/L sodium bicarbonate, 1% penicillin-streptomycin, and at 37 °C in a humidified atmosphere of 5% CO_2_/95% air.

### 2.8. Cytotoxicity of Extracts on 3T3-L1 Pre-Adipocytes

Cytotoxicity of the PTP and herb fruit extracts were measured by 3-(4,5-dimethylthiazol-2-yl)-2,5-diphenyl tetrazolium bromide (MTT) assay [33,34]. 3T3-L1 pre-adipocytes were seeded in 96-well plates at 10^4^ cells/well and incubated in medium (100 µL) at 37 °C in a humidified atmosphere of 5% CO_2_/95% air. After 24 h, the medium was replaced by samples (10–500 µg/mL in medium) for 24 h. Then, 50 µL of MTT working solution (1 mg/mL in PBS) was added to each well and incubated for a further 3 h at 37 °C under 5% CO_2_/95% air. Subsequently, the medium was removed, and 100 µL of DMSO was added to each well to dissolve the formazan crystals. The absorbance of the resultant formazan in a complete medium was measured at 595 nm using a microplate reader (Hybrid Multi-Mode detection Synergy H1, model H1MF, Bio-TeK Instruments, Santa Clara, CA, USA). All samples were conducted in triplicate.

### 2.9. Adipogenesis

The extracts influence adipocyte handling of an excess metabolic substrate by: (i) inhibiting glucose uptake and storage as triglycerides in proliferating and differentiated adipocytes, ‘adipogenesis’, and/or (ii) by such loaded adipocytes mobilizing or releasing their triglyceride stores, e.g., via lipolysis.

#### 2.9.1. Inhibition of Adipogenesis

3T3-L1 pre-adipocytes were seeded into 96-well plates at 2 × 10^3^ cells/well and incubated in DMEM/10% FCS and 50 U/mL of penicillin-streptomycin. At 100% confluence (day 0), the medium was replaced with one containing 0.5 mM IBMX, 1 µM dexamethasone, and 5 µg/mL insulin (Figure 1). On day 3, the medium was replaced every 2 days (days 3, 5, 7, 9) with DMEM/FBS/insulin and 100–500 µg/mL extract. On day 10, the cells were washed with PBS and stained with Oil Red O (Figure 1).

#### 2.9.2. Adipocyte Lipolysis

The protocol was identical to that above, except that extracts were only present between days 9–10 (Figure 1).

### 2.10. Quantification of Cellular Lipid Contents

At day 10, Oil-Red O-stained cells were fixed with 10% formalin for 1 h and incubated with staining solution at room temperature for 45 min. The cells were washed with 60% isopropanol/water 3-fold to remove the excess stain. Cell-accumulated Oil Red O dye was extracted using 100% isopropanol, and absorbance was measured at 500 nm by a microplate reader [35,36]. Caffeine and adrenaline were used as positive controls. All samples were determined in triplicate.

### 2.11. Chemical Profiling of PTP by LC/MS

The contents of aqueous extract of PTP water were identified and quantitated by liquid chromatography (LC) coupled to an electrospray ionization quadrupole time-of-flight mass spectrometer (LC-ESI-QTOF-MS/MS). The analysis conditions were detailed previously [37]. Briefly, separation used an Agilent 1260 Infinity Series HPLC System (Agilent Technologies, Waldbronn, Germany) with a Luna C-18 (2) column, 4.6 × 150 mm, 5 μm (Phenomenex Inc., Torrance, CA, USA) at 35 °C. The mobile phase was water (A) and acetonitrile (B), both with 0.1% (*v*/*v*) formic acid for ionization enhancement. The separation was via a 5–95% linear gradient of A over 0–30 min, then a 10 min hold, a flow rate of 0.5 mL/min, and injection volume of 10 μL. The mass detection was carried out on a 6540 Ultra-High-Definition Accurate Q–TOF-mass spectrometer (Agilent Technologies, Singapore) and was operated with electrospray ionization (ESI) in both positive and negative ion modes in the m/z range of 100–1000 Da. The mass spectrometric conditions were set as follows: nitrogen gas (N_2_) at 7 L/min and at 350 °C, nebulizer gas pressure at 30 psi capillary voltage 3.5 kV, fragmentation at 100 V, skimmer 65 V, Vcap 3500 V, octopole RFP 750 V. The collision energy was set at 10, 20 and 40 V using ultra-pure nitrogen gas (99.9995%). All mass acquisitions used Agilent MassHunter Data Acquisition Software, Version B.05.01 and analysis used Agilent MassHunter Qualitative Analysis Software B 06.0 (Agilent Technologies, Santa Clara, CA, USA). The m/z and fragmentation patterns of each compound were identified using metabolite databases such as the METLIN PCD/PCDL database (Agilent Technologies, Santa Clara, CA, USA), the public database Human Metabolome Database (http://www.hmdb.ca, accessed on 17 March 2022) and Lipid Maps (https://www.lipidmaps.org, accessed on 17 March 2022).

### 2.12. Statistical Analysis

All experiments were performed in triplicate. The data were displayed as means ± SDs or SEMs as indicated. Statistical analysis was performed using GraphPad Prism Version 8.0.1(San Diego, CA, USA) with one-way ANOVA. Differences with *p* < 0.05 were considered as significant.

## 3. Results and Discussion

### 3.1. Total Phenolic Contents

The phenolic contents using Folin-Ciocalteu’s reagent varied between 12.3 ± 0.6 and 108.4 ± 0.6 mg of GAE/g extract. Water clearly extracted greater amounts of phenolics than ethanol for all four crude preparations (Table 1).

### 3.2. Total Flavonoid Contents

Flavonoid contents were similar for the eight extractions, only varying between 0.80 ± 0.02 and 1.19 ± 0.09 mg QE/g extract. Water yielded marginally better extraction with three of the crude herbal preparations and made no apparent difference for *M. citrifolia* (Table 1).

### 3.3. DPPH Radical Scavenging

The DPPH assay is based on the ability of compounds in the sample to donate H to DPPH^•^ radicals. Aqueous and ethanolic extracts of *A. marmelos*, *M. citrifolia*, *C. sativum,* and PTP and positive controls, L-ascorbate acid and Trolox (Table 2).

In this study, three groups of nutritional compounds that contribute to general good health, i.e., phenolics, flavonoids, and free-radical scavengers. We focused on factors influencing obesity that independently contribute to metabolic disease. There are several trends that emerged from this study:

Aqueous extraction was superior to ethanol: for phenolics, flavonoids, H^•^ donation to radicals, inhibition of adipogenesis, and increased lipogenesis. This accords with previous research on total phenolic and flavonoid contents and a similar extraction selectivity for *A. marmelos* [17,31]. Other studies found that total phenolics in water and ethanolic extracts of *A. marmelos* (fruit) were 49.0, and 13.3 mg GAE/g extract and the total flavonoids were 11.9, and 4.4 mg QE/g extract [17]. The greater water solubility also accords with PTP sometimes being consumed as a solution. *A. marmelos* had substantial amounts of phenolic contents, but not the other 2 herbals susbtances, dilute out phenolics in PTP. For both extracts, the radical scavenging had the same trend, with *A. marmelos* being the most effective followed by PTP.

### 3.4. Cell Viability

Aqueous and ethanolic extracts of PTP and the three herb extracts showed no suggestion of any cytotoxicity on 3T3-L1 pre-adipocytes using the MTT assay (Figure 2), allowing their further study. Traditional medicine is based on synergy and synergistic effects and nullifying toxicity [38,39].

### 3.5. The Extracts Reduce Adipogenesis

#### 3.5.1. Effect on Lipid Accumulation

Both caffeine 53% of lipid and adrenaline 42% of lipid robustly reduced accumulated fat contents (Figure 3A). The four aqueous extracts more modestly but consistently and concentration-dependently also reduced fat accumulation by 28–31% for 500 ug/mL *A. marmelos*, *M. citrifolia*, and PTP, but by less for *M. citrifolia* (Figure 3A). Alcoholic extracts also concentration-dependently inhibited adipogenesis but consistently less so than for aqueous extracts. Markers for beta-oxidation were not measured, but our protocol was identical to studies demonstrating the same fate of fatty acids. Furthermore, insulin, which inhibits lipolysis, was present throughout. Thus, we surmise that lipid decline was, as least in part, due to lipid beta-oxidation [40].

#### 3.5.2. Lipolytic Actions

In just 24 h, 26 and 16% of lipid were lost with caffeine and adrenaline treatments, respectively (Figure 3B). Caffeine has several cellular actions but at the low concentration used here (10 uM), it blocks adipocyte adenosine A1 and A2A receptors at sub-micromolar concentrations [41,42]. Adrenaline also promotes lipolysis mainly via beta-1 and -2 receptors [43]. All four aqueous extracts appeared to have promoted lipid loss, effects suggestive of being concentration-dependent (Figure 3B). For ethanolic extracts, lipid losses were smaller. At 100 ug/mL, there was little overall consistent effect for all eight extracts. Aqueous extracts, *A. marmelos* and PTP had similar effects on both adipocyte responses, while *C. sativum* was marginally less effective and *M. citrifolia* was ambiguous.

Another potential antiobesity strategy targets brown fat cells containing abundant mitochondria that express the decoupling protein (UCP1). UCP1 switches free fatty acid beta-oxidation to heat production instead of ATP. Brown fat cells descend from the muscle cell line but are absent in obese patients. To circumvent this problem, a promising approach would be ‘browning’ of adipocytes from the preadipocyte lineage into beige adipocytes) that have more mitochondria and express UCP1 more than white adipocytes. These cells also express β3-adrenergic receptors (β3AR) that mediate thermogenesis and are only expressed in fat and urinary tract cells. Thus, β3AR poses a relatively specific drug target. 3T3-L1 cells were differentiated into the beige phenotype by the phenolic, sinapic acid using a similar protocol to ours [44]. Trigonelline from coffee causes 3T3-L1 cell browning by upregulating a β3AR/PKA/p38MAPK pathway, again using a protocol like ours. This pathway activates key mitochondriogenic genes, including UCP1 and lipolytic enzymes releasing fatty acids. But AMPK/SIRT1/PGC1α or mTORC1/COX-2 signaling pathways can brown 3T3-L1 adipocytes [45]. In animal studies, 18 other phytochemicals at 0.01–1% in the diet produce browning adipocytes and UCP1 elevation, weight loss and upregulate other beige fat cell markers [46]. Many studies in vitro using mostly 3T3-L1 adipocytes show that these phytochemicals are ligands for β3ARs, TRPV1, TRPM8, and TGR5 membrane sites coupled to the signaling pathway proteins that mediate signaling between the plasma membrane, mitochondria and lipid droplets [46].

While this in vitro data provides some proof of concept, applying it in vivo, especially in obese humans bearing refractory adipocytes, is an extreme challenge. There are some disconnects between isolated cells and clinically obese patients whose adipocytes are overinflated with lipids, inflamed, and refractory. The diverse range of polyphenols and flavonoids has poor bioavailability, commonly yielding nanomolar plasma concentrations. Yet, they have convincing effects in animals while in vitro actions mechanisms mid-micromolar concentrations to reveal effects. A possible explanation for these discrepancies is that polyphenols, especially flavonoids, are partially metabolized by intestinal bacterial creating a smaller number of mostly well-absorbed monocyclic polyphenols, thereby achieving higher blood concentrations [47]. Accordingly, human trials may have erratic outcomes. In addition, any medication requires an anti-inflammatory against the M1 macrophages that are largely responsible for this condition. Inflammation is a generic process, but cellular and molecular mediators vary by etiology and body site [48].

Nutrient-based anti-inflammatories appear to have a wide range of actions covering different scenarios [49] and are likely able to target adipocyte inflammation with their own set of unique adipokines. Long-chain ω-3 polyunsaturated fatty acids reduce inflammatory cytokines and produce protectins and resolvins, lipids that induce recovery from inflammation [50]. Removing long-chain saturated fatty acids that appear to promote adipose inflammation [51] from the diet may augment this effect.

A diet providing a negative energy balance supplemented with full nutrient needs is also needed. In vivo, resveratrol activates the β-adrenergic receptors, and oleuropein aglycone [52] and pentamethylquercetin [53], also increasing UCP1 expression. Chlorogenic acid stimulates brown adipocyte thermogenesis via increasing glucose uptake and mitochondrial function [54,55]. Ginsenoside Rg3 also browns 3T3-L1 adipocytes causes expression of UCP1 and related genes, and reduction of lipid droplets [56]. Since these authors appeared to use the same pre-adipocyte differentiation control to ours, the lipolysis group cells may have dissipated lipid oxidatively. Since insulin depresses lipolysis [57] and was present throughout our protocol, decoupled fatty acid oxidation may account for a significant proportion of lipid lost during days 9–10. Fatty acids spill over into the circulation risking systemic inflammation. Thus, future experiments should unravel the role of mitochondrial, uncoupling and optimizing its expression without inducing heat stress. Trigonelline reduces adipogenesis via influencing PPAR expression, which results in subsequent downregulation of the PPAR-mediated pathway during adipogenesis [58]. Moreover, trigonelline’s browning impact in 3T3-L1 white adipocytes is achieved via activating 3-AR and inhibiting PDE4, consequently stimulating the p38 MAPK/ATF-2 signaling pathway, according to mechanistic studies [45]. Capsaicin generates lipolytic action in adipocytes by increasing triacylglycerol hydrolysis, and these effects are mediated at least in part by regulation of the expression of multiple genes involved in the lipid catabolic pathway, including HSL and CPT-I, as well as those involved in thermogenesis, which including UCP2 [59]. Inhibiting 3T3-L1 preadipocyte development may indeed be a potential therapeutic strategy for obesity. This prompted us to investigate the anti-adipogenic effects of a mixture of the fruits of the three herbs, *A. marmelos*, *M. citrifolia*, and *C. sativum*, in the 3T3-L1 cell line. It has been observed that the PTP and its components significantly reduce lipid formation in differentiated 3T3-L1 adipocytes. In our study, water extracts reduced lipid deposition more significantly than ethanol extracts in differentiated 3T3-L1 cells. Therefore, we decided to examine the active component of the water extracts based on this discovery. In the current investigation, PTP and their extracts were found to suppress cell differentiation and lipid accumulation in cells such as adipocytes via target signal transduction.

We hypothesize that the phytochemicals in PTP changed the lipogenesis pathway in 3T3-L1 preadipocytes. In the future, an evaluation of the mechanism of the action of this combination extract from PTP is suggested to study its expression of vital adipogenic proteins and genes. Further research is recommended to determine their efficacy and safety in animal and clinical models.

### 3.6. Structural Elucidation of PTP by LC/MS

Phytochemical screening for secondary metabolites in the PTP combination of *A. marmelos*, *M. citrifolia*, and *C. sativum* was characterized by LCESI-QTOF-MS/MS in both negative and positive ion modes. Their structures were inferred from their mass spectra and fragmentation patterns.

These compounds were identified by accurate mass measurement (mass error) that were obtained from the observed mass of their protonated [M + H]^+^ and deprotonated [M − H]^−^, or other adduct ions, compared to the theoretical exact mass in databases. The fragment patterns (MS/MS) were also used to confirm the structure of these compounds by their unique fragmentation characteristics. Table 3 lists the negative and positive molecular ions, along with their fragmented ions and tentative identification of the compounds which showed 67 compounds. The total ion current (TIC) chromatograms of negative and positive mode LC-ESI-QTOF-MS/MS are shown in Figure 4 and Appendix A.

For the phytochemical profiles by LC-ESI-QTOF-MS/MS, the three herbal fruits/seed aqueous extracts had diverse signatures in their contents. *M. citrifolia* had large peaks of predominantly high MWs at 5–16 min, while *A. marmelos* compounds eluted over 9–13 min. These two extracts contributed most to PTP while *C. sativum* extract added little to PTP (Figure 4). Eluted material by ESI positive mode tended to reflect ESI negative signals.

Compounds comprised acids, alcohols, carotenoids, esters, flavonoids, ketones, lactones, lignans, nucleosides, quinones, phenols, saccharides, triterpenoids, iridoids, triterpenoids, aromatic, or glycosides thereof (Table 3). Of these, phenols, flavonoids, alkaloids, and terpenoids are commonly bioactive in vitro [40,60], give useful information regarding their anti-obesity activities, which include suppressing white adipogenesis, lipogenesis, and increasing lipolysis and muscle thermogenesis.

In the PTP formula, five iridoid compounds—menotropin (17), deacetyl asperulosidic acid (20), asperulosidic acid (29), asperuloside (30) and phlomiol (40)—were found. The first four compounds (17, 20, 29, and 30) were found in the *M. citrifolia* fruit and phlomiol (4) was found in *C. savitum*. The flavonoid glycosides—sakuranetin (22) isosakuranetin (23) luteolin 7-(6″-p-benzoyglucoside) (28), 4′-hydroxy-5,7,2′-trimethoxyflavanone 4′-rhamnosyl-(1->6)-glucoside (36) rhamnazin 3-rutinoside (38) and rutin (46)—were also found in *M. citrofolia* fruit. Acacetin 7-(4″″-acetylrhamnosyl)-(1-6)-glucosyl-(1-3)-(6″-acetylglucoside) (31, 35), acacetin 7-(2″-acetylglucoside) (32), eriodictyol 7-O-glucoside (43), and 5,7,8-trihydroxyflavanone 7-saglucoside (50) were flavonoid glycosides from *A. marmelos*. Isorhamnetin (24) was detected in *C. savitum*. Terpene pteroside D (27) was found in *M. citrofolia* and sanshiside D (39,42) was detected in *A. marmelos*. More details are in Table 3. Most Thai traditional prescriptions contain multiple herbal ingredients, but *A. marmelos* alone might satisfy the therapeutic and nutritional treatment. Future studies should explore the value, if any, of contributions of other components to the overall therapeutic outcomes.

Flavonoids suppress adipocyte differentiation and inhibit fat lipid via transcription factors including C/EBPα, C/EBPβ, PPARγ, and SREBP-1c and activation of AMPK [61,62,63,64,65]. In the same fashion as our work, chlorogenic acid was identified in the deprotonated molecular ion [M − H]^−^, which was found in *A. marmelos*. At physiological and supraphysiological concentrations, isorhamnetin, a metabolite of quercetin, prevented the differentiation of 3T3-L1 pre-adipocytes to adipocytes. Isorhamnetin was more efficient than quercetin in suppressing fat accumulation in 3T3-L1 pre-adipocytes at physiologically attainable doses [66,67]. Also, we discovered isorhamnetin in the deprotonated molecular ion [M − H]^−^ observed in *C. sativum* [68,69].

In general, antioxidant activity is determined by the quantity and location of hydroxyl groups and other substituents, as well as the glycosylation of flavonoid molecules. Furthermore, anti-adipogenesis activity revealed that flavonols of the methoxy group at the 3-position had the most potent anti-adipogenic action. Similarly, the location of methoxy groups in the B-ring correlates to the anti-adipogenic effect of flavonols. In our findings, when the IC_50_ values of individual herbs (*A. marmelos*, *M. citrifolia*, and *C. sativum*) were compared, flavonols having two and three hydroxyl groups in B-rings showed the strongest antioxidant activity [70,71,72].

In Figure 4, the deprotonated molecular ion [M − H]^−^ of peak no. 36 stands out as the flavonoid group in PTP, that was discovered in *M. citrifolia* to be 3,4,7-trihydroxy-5-methoxy-8-prenylflavan4-O-(beta-D-xylopyranosyl-(1->6)-beta-D-glucopyranoside). This compound contained a hydroxyl group at the 3-position of the C-ring, which was linked to the anti-differentiation action of adipocyte and antioxidant activity. The findings in our study are evidence that quercetin and rutin are also found in *M. citrifolia*, and *C. sativum*. This supports the findings of [65,73], that the high nutritional value of *A. marmelos*, *M. citrifolia*, *C. sativum* and PTP may induce therapeutic effects, including antioxidant and anti-adipogenesis.

## 4. Conclusions

The results showed that the extracts of PTP and the individual herbs contained a certain amount of flavonoid and phenolic components which suggests antioxidant potential. The ability of PTP extracts to decrease 3T3-L1 adipocyte differentiation was observed. We conclude that PTP water and ethanol extracts reduce lipid accumulation in differentiated 3T3-L1 cells and prevent adipogenesis at higher dosages, as demonstrated by the Oil Red O assay. The LC-ESI-QTOF-MS/MS investigation of the PTP profile and individual herbs reveals that the three herbs (fruits) extracts include polyphenols, which may explain the PTP water extract’s antioxidant and anti-adipogenic activities. The active compounds found in the PTP water extract include malic, quinic, chlorogenic acids, quercetin, rutin, isorhamnetin and others.

It has the potential to be utilized as a supplement to boost the nutritional value of food. Further work using animals and clinical studies may yield an effective food-based treatment for the amelioration of obesity and metabolic disease.

## Figures and Tables

**Figure 1 foods-11-03067-f001:**
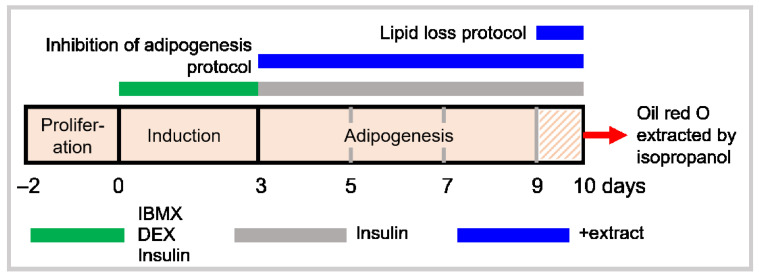
A schema outlining the two protocols. 3T3-L1 cell differentiation into adipocytes was induced by stimulation with hormones dexamethasone (DEX), insulin, and isobutylmethylxanthine (IBMX).

**Figure 2 foods-11-03067-f002:**
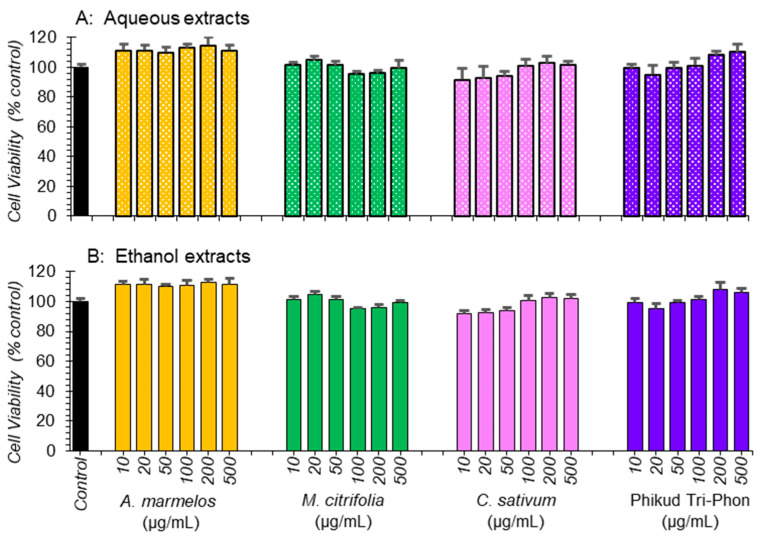
Cytotoxicity of (**A**) aqueous and (**B**) ethanol extracts of *A. marmelos, M. citrifolia*, *C. sativum*, and PTP water (**A**) and ethanol extracts (**B**) testing the viability of 3T3-L1 cells. Cytotoxicity was measured using the MTT assay, and the results are expressed as means ± SEM (*n* = 3). Responses are expressed as % of corresponding controls.

**Figure 3 foods-11-03067-f003:**
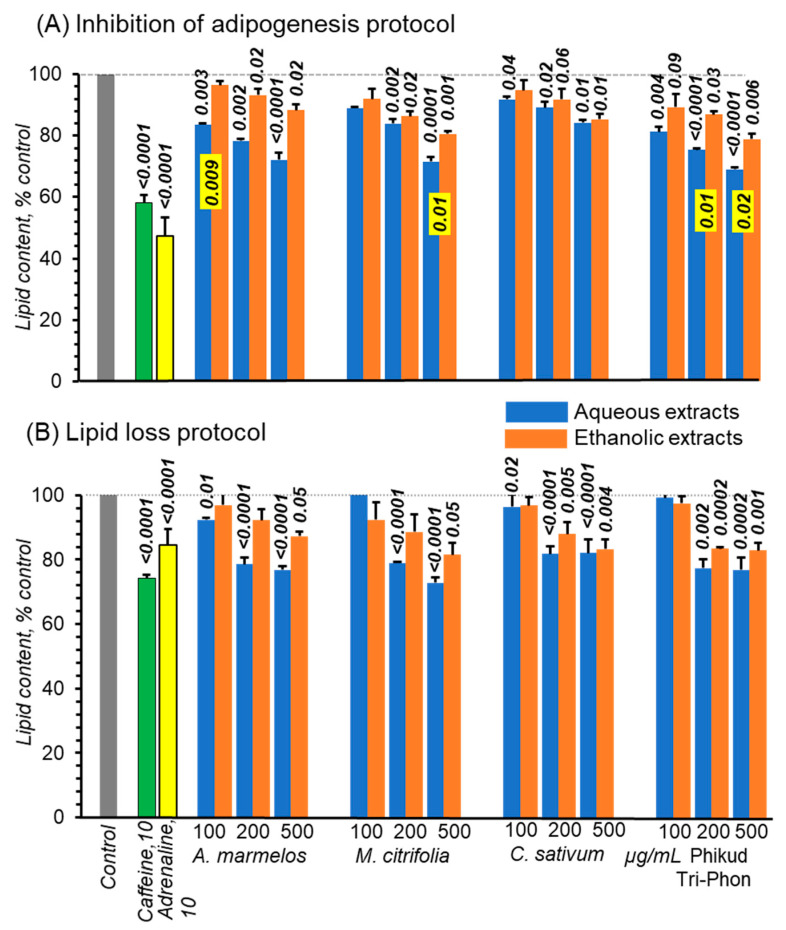
Lipid accumulation in 3T3-L1 cells after 7 days of adipogenesis. In (**A**), the cells were also treated with, either one of four aqueous extracts at 100–500 ug/mL of extract indicated (Blue bars) or ethanolic extracts from the same preparations (Orange bars). For (**B**), the protocol was identical except the same series of extracts were only present for day 7 and that represents loss of accumulated triglycerides. All concentrations are µg/mL, error bars are means ± SD, *n* = 3 and *p*-values comparing control by one-way ANOVA, *p*-values in yellow boxes compare differences between aqueous and ethanolic extracts by 2-tailed Student’s *t*-test.

**Figure 4 foods-11-03067-f004:**
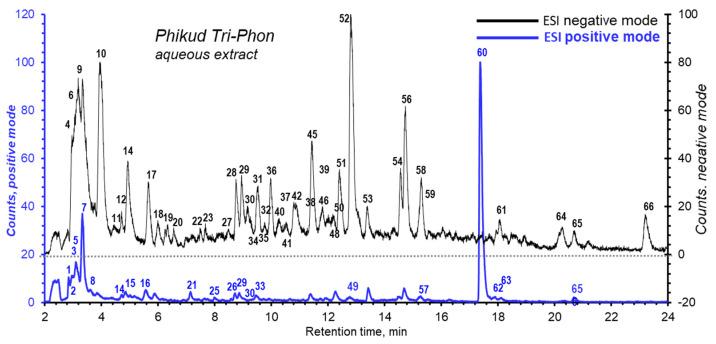
Total ion chromatogram of 10 mg/mL aqueous extract of PTP by LC/MS operated in ESI negative mode and in ESI positive mode. The characteristics of each peak are in Table 3.

**Table 1 foods-11-03067-t001:** Total phenolic content (mg GAE/g extract) and flavonoid content (mg QE/g extract).

Samples	Total Phenolic Content (mg GAE/g Extract)		Total Flavonoid Content (mg QE/g Extract)	
	Aqueous Extract	Ethanolic Extract	Aqueous Extract	Ethanolic Extract
*A. marmelos*	108.4 ± 0.6	99.5 ± 0.2	1.02 ± 0.01	0.80 ± 0.02
*M. citrifolia*	21.8 ± 0.4	12.3 ± 0.6	1.01 ± 0.07	1.12 ± 0.13
*C. sativum*	22.3 ± 0.7	18.4 ± 0.4	0.93 ± 0.03	0.82 ± 0.03
PTP	62.4 ± 0.5	49.3 ± 0.7	1.19 ± 0.09	0.98 ± 0.07

Data are presented as means ± SD (*n* = 3). PTP = Phikud Tri-Phon.

**Table 2 foods-11-03067-t002:** DPPH radical scavenging of *A. marmelos*, *M. citrifolia*, *C. sativum* and PTP extracts from aqueous and ethanol extractions.

Extract/Sample	IC_50_ (µg/mL)	
	Aqueous Extract	Ethanolic Extract
*A. marmelos*	79.1 ± 6.0	201.7 ± 7.8
*M. citrifolia*	181.7 ± 6.4	>2000
*C. sativum*	139.4 ± 12.7.	>2000
PTP	92.4 ± 6.6	292.1 ± 32.7
L-ascorbic acid	7.2 ± 0.1	
Trolox	11.3 ± 0.7	

Data are presented as means ± SD (*n* = 3).

**Table 3 foods-11-03067-t003:** Tentative identification of compounds found in Tri-Phon (+/−ESI), *A. marmelos* (A), *M. citrifolia* (M), and *C. sativum* (C).

No.	RT (min)	*m/z*	Adduct	MS/MS	Tentative Identification	Formula	Error(ppm)	Plant	Group of Compounds
1	2.855	104.1064	[M]^+^	60.0798, 58.0640	Choline	C_5_H_14_NO	10.94	AMC	Amino acid
2	3.109	118.0853	[M]^+^	58.0641	Betaine	C_5_H_12_NO_2_	12.73	A	Amino acid
3	3.117	160.0957	[M + H]^+^	88.0745, 58.0640	Isovaleryglycine	C_7_H_13_NO_3_	6.99	MC	Amino acid
4	3.133	195.0493	[M − H]^−^	161.0250, 129.0121, 85.0239, 75.0043, 59.0096	Gluconic acid	C_6_H_12_O_7_	8.85	C	Monosaccharides
5	3.187	116.0696	[M + H]^+^	70.0640, 58.0640	Proline	C_5_H_9_NO_2_	8.66	A	Amino acid
6	3.219	439.0899	[M + CH3COO]^−^	337.0640, 337.6055, 96.9641, 78.9543	4-Methoxybenxyl O-(2-sulfoglucoside)	C_14_H_20_O_10_S	3.8	AMC	sulfoglucoside
7	3.338	144.1007	[M + H]^+^	84.0798, 58.0641	L-2-Amino-3-methylenehexanoic acid	C_7_H_13_NO_2_	8.36	AMC	Amino acid
8	3.388	158.1164	[M + H]^+^	98.0939, 70.0640, 58.0640	3-(piperidin-3-yl)propanoic acid	C_8_H_15_NO_2_	7.31	MC	Amino acid
9	3.364	191.0572	[M − H]^−^	93.0292, 85.0246, 71.0095, 59.0101	Quinic acid	C_7_H_12_O_6_	−5.7	A	acid
10	3.981	133.0115	[M − H]^−^	114.9976, 71.0094	Malic acid	C_4_H_6_O_5_	20.65	AMC	acid
11	4.477	189.0057	[M − H]^−^	99.0028, 83.0086, 55.0149	Oxalosuccinic acid	C_6_H_6_O_7_	−8.59	M	acid
12	4.721	290.0905	[M-H2O-H]^−^	128.0272	Glucose-6-glutamate	C_11_H_19_NO_9_	−8.14	M	Amino sugar
13	4.804	203.0208	[M − H]^−^	79.0130, 71.0089	Oxaloglutarate	C_7_H_8_O_7_	−5.29	A	acid
14	4.849	193.0328	[M + H]^+^	111.0006, 89.0553	Citric acid	C_6_H_8_O_7_	7.66	AC	acid
		191.0209	[M − H]^−^	111.0019, 87.0029	Citric acid	C_6_H_8_O_7_	−6.11	AC	acid
15	4.849	175.0227	[M + H]^+^	133.0226, 111.0007	Dehydroascorbic acid	C_6_H_6_O_6_	5.8	A	acid
16	5.587	294.151	[M + H]^+^	276.1413, 258.1299, 230.1364, 144.100, 86.0950	*N*-(1-Deoxy-1-fructosyl)isoleucine	C_12_H_23_NO_7_	12.68	MC	Amino sugar
17	5.67	389.1138	[M − H]^−^	209.0352, 137.0526, 89.0178, 59.0092	Menotropein	C_16_H_22_O_11_	−12.5	M	Iridoid
18	6.056	117.0195	[M − H]^−^	73.0242	Succinic acid	C4H6O4	−1.43	C	acid
19	6.252	530.1593	[M − H]^−^	504.9040, 206.0347, 162.0470, 160.0313	Unidentified			C	
20	6.565	389.1151	[M − H]^−^	350.6325, 330.9792, 278.4458, 210.0532, 139.0295, 100.0060, 71.0070	Deacetyl asperulosidic acid	C_16_H_22_O_11_	−12.5	M	Iridoid
21	7.158	166.0848	[M + H]^+^	120.0791, 103.0530, 77.0376	Phenylalanine	C_9_H_11_NO_2_	9.97	AMC	Amino acid
22	7.513	447.1354	[M − H]^−^	179.0415	Sakuranetin	C_22_H_24_O_10_	−12.81	M	Flavonoid glycosides
23	7.676	447.1346	[M − H]^−^	387.2598	Isosakuranin	C_22_H_24_O_10_	−11.02	M	Flavonoid glycosides
24	8.08	315.0552	[M − H]^−^	152.0007, 108.0132	Isorhamnetin	C_16_H_12_O_7_	−13.25	C	Flavonoids
25	8.605	188.069	[M]^+^	146.0579, 118.0634	*N*-(2,5-dihydroxyphenyl)pyridinium	C_11_H_10_NO_2_	11.45	A	Hydroquinones
26	8.713	193.0484	[M + H]+	137.0568, 89.0378	Scopoletin	C_10_H_8_O_4_	5.88	M	coumarin
27	8.765	445.1571	[M + Cl]^−^	409.1380, 361.7736, 179.0421, 59.0077	Pteroside D	C_21_H_30_O_8_	14.31	M	Terpene
28	8.813	551.1296	[M − H]^−^	-	Luteolin 7-(6″-p-benzoyglucoside)	C_28_H_24_O_12_	−18.33	M	Flavonoid glycosides
29	8.837	450.1564	[M + NH4]^+^	304.1289, 235.0563, 193.0479, 147.0416	Asperulosidic acid	C_18_H_24_O_12_	9.33	M	Iridoid glycoside
		431.1285	[M − H]^−^	251.0376, 165.0430, 89.0167, 59.0081	Asperulosidic acid	C_18_H_24_O_12_	−20.88	M	Iridoid glycoside
30	8.837	415.1223	[M + H]+	235.0548, 147.0396	Asperuloside	C_18_H_22_O_11_	2.86	MC	Iridoid glycoside
31	9.162	883.2506	[M + HCOO]^−^	799.2357, 750.1742, 672.1283, 568.0859, 450.6009, 343.1243, 233.1087, 176.6268, 98.5898	Acacetin 7-(4″″-Acetylrhamnosyl)-(1-6)-glucosyl-(1-3)-(6″-acetylglucoside)	C_38_H_46_O_21_	0.86	A	Flavonoid glycosides
32	9.189	487.111	[M − H]^−^	271.0498, 153.0092	Acacetin 7-(2″-acetylglucoside)	C_24_H_24_O_11_	27.89	A	Flavonoid glycosides
33	9.477	414.193	[M + NH4]^+^	259.0775, 163.0569, 133.0478	Unidentified			MC	
34	9.509	415.1452	[M + Cl]^−^	357.1600, 159.7679	Prenyl arabinosyl-(1->6)-glucoside	C_16_H_28_O_10_	−18.19	M	Fatty acyl glycosides
35	9.536	883.2502	[M + HCOO]^−^	567.0918, 387.0325, 259.0462, 165.0066	Acacetin 7-(4″″-Acetylrhamnosyl)-(1-6)-glucosyl-(1-3)-(6″-acetylglucoside)	C_38_H_46_O_21_	1.31	A	Flavonoid glycosides
36	9.983	637.2244	[M − H]^−^	548.1696, 505.7741, 443.1224, 361.1271, 277.1135, 221.0513, 179.0444, 89.0174, 89.0171	4′-Hydroxy-5,7,2′-trimethoxyflavanone 4′-rhamnosyl-(1->6)-glucoside	C_30_H_38_O_15_	−16.64	M	Flavonoid glycosides
37	10.004	353.0935	[M − H]^−^	191.0444, 179.9262	Caffeoyl quinic acid	C_16_H_18_O_9_	−16.13	AC	Phenolic acid
38		637.1844	[M − H]^−^	221.0513, 179.0444, 89.0174	Rhamnazin 3-rutinoside	C_29_H_34_O_16_	−10.97	M	Flavonoid glycosides
39	10.261	867.2521	[M − H]^−^	705.1583, 551.0980, 417.0741, 271.0386, 190.9786, 125.0144	Sanshiside D	C_39_H_48_O_22_	5.01	A	Iridoid glycoside
40	10.307	437.1288	[M − H]^−^	386.2684, 197.2444	Phlomiol	C_17_H_26_O_13_	2.89	C	Iridoid glycoside
41		385.0752	[M − H]^−^	285.7925, 191.0090, 85.0225	Feruloylglucaric acid	C_16_H_18_O_11_	6.31	A	Phenolic acid
42	10.798	867.2543	[M − H]^−^	705.1604, 551.0937, 389.0427, 311.0391, 125.0158	Sanshiside D	C_39_H_48_O_22_	2.48	A	Iridoid glycoside
43	10.911	449.1176	[M − H]^−^	287.0413, 269.0311, 259.0470, 125.0154	Eriodictyol 7-O-glucoside	C_21_H_22_O_11_	−19.29	A	Flavonoid glycosides
44	11.394	325.1122	[M + H]^+^	163.0597, 85.0275	Moracin L	C_19_H_16_O_5_	−15.84	MC	Carbonic acids
		458.2196	[M + NH4]^+^		Diferuloylputrescine	C_24_H_28_N_2_O_6_			Phenolic amide
45	11.43	475.1682	[M + Cl]^−^	323.0847, 263.0568, 179.0450, 115.0669, 79.0121, 59.0075	Diferuloylputrescine	C_24_H_28_N_2_O_6_	−8.55	M	Phenolic amide
46	11.693	611.1559	[M + H]^+^	576.4255, 465.0904, 303.0463, 147.0633	Rutin	C_27_H_30_O_16_	7.79	M	Flavonoid glycosides
46	11.828	609.158	[M − H]^−^	300.0108, 271.0113, 255.0165, 150.9935, 107.0051, 63.0188	Rutin	C_27_H_30_O_16_	−19.52	M	Flavonoid glycosides
47	11.971	273.0731	[M + H]^+^		Maracin J	C_15_H_12_O_5_	9.7	A	
48	12.041	367.1097	[M − H]^−^	173.0343, 93.0267	Feruloylquinic acid	C_17_H_20_O_9_	−17.01	A	Phenolic acid
49	12.274	453.3406	[M + H]^+^	435.3290, 376.0758, 336.2258, 285.1264, 245.1782, 210.1452, 175.0743, 139.0846, 100.0526, 55.0526	3-Oxo-12,18-ursadien-28-oic acid	C_30_H_44_O_3_	−9.44	MC	Triterpenoids
50	12.394	433.1226	[M − H]^−^	271.0476, 150.9942, 107.0083	5,7,8-Trihydroxyflavanone 7-glucoside	C_21_H_22_O_10_	−19.81	A	Flavonoid glycosides
51	12.431	851.2593	[M − H]^−^	689.1579, 563.1371, 401.0850, 325.5702, 255.0522, 125.0147	Peracetylmacrophylloside D	C_39_H_48_O_21_	2.62	A	
52	12.772	648.3033	[M + NH4]^+−^	325.1082, 289.0870, 163.0578	Nonioside B	C_26_H_46_O_17_	6.21	M	oligosaccharides
52	12.826	665.2575	[M + Cl]^−^	485.1337, 389.1604, 305.1470, 179.0456, 89.0173	Nonioside B	C_26_H_46_O_17_	−21.94	M	oligosaccharides
53	13.392	713.4895	[M + Cl]^−^	654.4719, 601.0903, 558.4126, 488.5905, 427.7981, 318.4760, 258.7616, 203.0478, 136.2926, 73.9288	Bis{2-[2-(dodecyloxy)ethoxy]ethyl} benzene-1,2-dicarboxylate	C_40_H_70_O_8_	8.9	MC	
53	13.431	679.5083	[M + H]^+^	552.4446, 436.3200, 336.2268, 210.1470, 100.1106	Bis{2-[2-(dodecyloxy)ethoxy]ethyl} benzene-1,2-dicarboxylate	C_40_H_70_O_8_	8.9	MC	
54	14.569	503.2024	[M + Cl]^−^	263.0625, 143.0988, 59.0078	Unidentified	55		M	
55	14.677	486.251	[M + NH4]^+^	325.1078, 163.0576, 85.0272	Unidentified			MC	
55		325.1119	-	85.0263	Moracin derivative	C_19_H_16_O_5_	−14.92	MC	
56	14.736	503.2014	-	389.1673, 323.0829, 263.0625, 143.0989, 89.0177, 59.0082	Unidentified		−18.14	M	
57	15.267	289.163	-	127.1096, 69.0321	Unidentified			MA	
58	15.274	503.2008	-	179.0436, 143.0983, 113.0153, 89.0179, 59.0073	Unidentified			M	
59	15.297	693.2956	[M + Cl]^−^	333.1779, 221.0549, 179.0453, 119.0263, 89.0173	Methylcellulose	C_29_H_54_O_16_	21.62	M	
60	17.388	274.2728	[M + H]^+^	256.2618, 106.0849, 88.0747, 70.0643, 57.0689	Hexadecasphinganine	C_16_H_35_NO_2_	4.58	AMC	sphingoid
61	18.065	763.3356	[M − H]^−^	619.1710, 403.2154, 305.1434, 277.1116, 179.0436, 115.0680	Unidentified			M	
62	18.097	327.1683	[M + H]^+^	259.1053, 241.0943, 198.0887, 135.0529, 106.0273, 69.0688, 51.0226	Unidentified			A	
63	18.840	244.2613	[M + H]^+^	226.2483, 76.0747, 58.0641	3-Lauryloxypropylamine	C_15_H_33_NO	8.97	AMC	N compound
64	20.253	791.3657	[M + Cl]^−^	691.1349, 611.8238, 529.0908, 431.2437, 305.1467, 233.1283, 179.0465, 101.0156, 89.0161	Unidentified			M	
65	20.68	286.2432	[M − H]^−^	200.0431, 146.4621, 88.0326	Prosopinine	C_16_H_33_NO_3_	−15.49	M	Alkaloid
65	20.708	288.2522	[M + H]^+^	242.2443, 88.0745	Prosopinine	C_16_H_33_NO_3_	3.89	AM	Alkaloid
66	23.223	293.1777	[M − H]^−^	236.0874, 221.1447, 205.1122, 148.0444, 107.0432	Phytuberin	C_17_H_26_O_4_	−6.37	MC	Sesquiterpenoids
67	28.584	307.1472	[M + H]^+^	291.2441, 258.8319, 238.0707, 210.0759, 183.0840, 133.0842, 106.0254, 79.0407, 52.0293	Unidentified			A	

## Data Availability

All raw data used for figure generation in this manuscript can be obtained by contacting the corresponding author.

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
