# Peer review of "A Thai Traditional Triple-Fruit Formulation “Phikud Tri-Phon” May Provide Fat Loss and Nutritional Benefits"

_foods, 2022, doi:10.3390/foods11193067_

Round 1
Reviewer 1 Report
The manuscript mainly discusses the nutritional and health benefits of triple fruit formulation. This is a good study, but some issues should be addressed. The authors have studied the formulation’s antioxidant and anti-adipogenic properties using 3T3-L 1 cells. Further, they conducted LC/MS-MS analysis.
1. Abstract – A sentence stating synergism should be added.
2. Emphasize the novelty of the study.
3. Give English names of the studied plants.
4. Include previous comprehensive studies conducted on sperm genotoxicity in the introduction section.
5. Indicate the whether the dried fruits, especially A. Mamalosa and M citrifolia are ripe or unripe. The phytochemical components may change at different stages.
6. The extraction procedure should be given briefly—especially the temperatures etc.
7. What were the concentration ranges used to determine phenolic and flavonoid contents?
8. Did the authors conduct different extract procedures for antioxidants using methanol?
9. How many replicates were taken for MTT and lipid assays?
10. The authors could have used another antioxidant assay to confirm the activity.
11. Why didn’t the authors measure Triglycerides content”? See doi:10.3390/biom10020165.
12. Section 3.5.2.- the authors discuss the percentage loss of lipids with adrenaline and caffeine treatments. The methodology did not reflect either of these treatments.
13. Figure 3 is too crowded. You can convert it to a table.
14. Figure 4 should accompany by a table. Give retention and a detailed table, at least in the appendix.
15. Discussion – “Aqueous extracts, A. marmelos and PTP had similar effects on both adipocyte responses, while C. sativum was marginally less effective, and M. citrifolia was ambiguous.” However, there could be a synergistic effect that is taking place. TM is based on synergistic and synergistic effects and nullifying toxicity. Therefore, the authors should conduct a detailed study before discussing the above.
Author Response
26th September 2022
Manuscript ID: foods-1899094
Response to the editors and reviewers of the Foods Journal
Dear Milena Petrovic Markovic,
Regarding reviewers’ comments and suggestions on manuscript entitled “A Thai Traditional triple fruit formulation “Phikud Tri-Phon” may provide fat loss and nutritional benefits” for publication in the Journal of Foods, we do appreciate and grateful for you and reviewer’s valuable statements to improve our manuscript. We have corrected according to all comments by using the track changes in the manuscript. We also wrote the responses for all comments point by points below.
Response to Reviewer 1 Comments
The manuscript mainly discusses the nutritional and health benefits of triple fruit formulation. This is a good study, but some issues should be addressed. The authors have studied the formulation’s antioxidant and anti-adipogenic properties using 3T3-L 1 cells. Further, they conducted LC/MS-MS analysis.
Thank you for taking the time to read our MS and for your comments
Comment 1: Abstract – A sentence stating synergism should be added.
Response 1: We have added in the abstract, line 24-25.
Comment 2: Emphasize the novelty of the study.
Response 2: We have added in the abstract, line 24-25.
Comment 3: Give English names of the studied plants.
Response 3: We have explained the English name of triple fruits and changed in the introduction, page 2, line 71-72.
Comment 4: Include previous comprehensive studies conducted on sperm genotoxicity in the introduction section.
Response 4: We have noted genotoxicity the introduction, line 75.
Comment 5: Indicate the whether the dried fruits, especially A. Mamalosa and M citrifolia are ripe or unripe. The phytochemical components may change at different stages.
Response 5: We have revised in Plant Materials section, page 3, line 114.
Comment 6: The extraction procedure should be given briefly—especially the temperatures etc.
Response 6: We have added the extraction methods of Phikud Tri-Phon (PTP) and the 3 herbal products, page 3, line 121.
Comment 7: What were the concentration ranges used to determine phenolic and flavonoid contents?
Response 7: We have explained in 2.4 Determination of Total Phenolic Contents of gallic acid solutions at 1.56-200 µg/mL, page 3, line 136 and 2.5 Determination of Total Flavonoid Contents of quercetin at 1.6-50 µg/mL, line 144. Also, concentration for the extract samples as 1 mg/mL, line 131 as indicated with the yellow highlight. The different of concentration ranges were showed the absorbance range between 0.2-0.8, which are suitable for used cal. Equivalent of the extracts.
Comment 8: Did the authors conduct different extract procedures for antioxidants using methanol?
Response 8: No, it is not different. Because of the blank of control, the selected solvent will provide the same result.
Comment 9: How many replicates were taken for MTT and lipid assays?
Response 9: In all our experiment, all experiments were in triplicate.
Comment 10: The authors could have used another antioxidant assay to confirm the activity.
Response 10: Total phenolics were found to be highly correlated with DPPH, suggesting that they contribute to the antioxidant properties of the plants studied. Another antioxidant assay has been reported in a previous study review (https://doi.org/10.3390/ijms231810889 , https://www.cabdirect.org/cabdirect/abstract/20113062991, and https://doi.org/10.1007/s13749-014-0064-8).
In another study, the methanolic extract of bael fruit was evaluated for its antioxidant activity via, DPPH and FRAP (ferric-reducing antioxidant power) assay. From the findings, it was determined that fruit extract show IC50 value of 52.06 µg/mL and 59.32 µmol/g for DPPH and FRAP assay, respectively. As a result, in this study, we only used the DPPH assay to screen for and confirm the activity range.
Comment 11: Why didn’t the authors measure Triglycerides content”? See doi:10.3390/biom10020165.
Response 11: Thank you very much for your question. In this study, we thought that the Oil Red O assay for anti-adipogenesis is enough to screen samples.
Comment 12: Section 3.5.2.- the authors discuss the percentage loss of lipids with adrenaline and caffeine treatments. The methodology did not reflect either of these treatments.
Response 12: We have added the information in the methodology section 2.10. Quantification of cellular lipid contents at line 202.
Comment 13: Figure 3 is too crowded. You can convert it to a table.
Response 13: Figure 3 shows all the p values and journals are increasingly requiring this and the figure is clearer.
Comment 14: Figure 4 should accompany by a table. Give retention and a detailed table, at least in the appendix.
Response 14: This is already in table S1, but we have been asked to move it to the main MS which we have done.
Comment 15: Discussion – “Aqueous extracts, A. marmelos and PTP had similar effects on both adipocyte responses, while C. sativum was marginally less effective, and M. citrifolia was ambiguous.” However, there could be a synergistic effect that is taking place. TM is based on synergistic and synergistic effects and nullifying toxicity. Therefore, the authors should conduct a detailed study before discussing the above.
Response 15: We have revised and mention in 3.4 cell viability section at line 268.
We had responded to all comments by the reviewers and made changes to the manuscript when it was needed. We hope that the paper is now acceptable for publishing in Foods.
Thank you very much
Yours Sincerely,
Sakan Warinhomhoun, Ph.D. (Corresponding author)
School of Medicine, Walailak University,
222 Thaiburi, Thasala district, Nakhon Si Thammarat 80160, Thailand
Tel.: +66 75677417 E-mail: Sakan.wa@wu.ac.th, sakan.cu@gmail.com

Reviewer 2 Report
Comments
The paper refers to the scientific field covered by the journal.
There are some typos and English needs revision in places.
Abstract: line 24: aimed „to investigate“, Line 25: compositions or better components; Line 28: extracts (plural) , ranged instead of were; Line 29 Trolox equivalents (TE), please define also the above used abbreviations (QE, GAE), etc.
The Abbreviations could be deleted from the abstract as it can stand independently. Also when the phrase is first time used it should be defined in the main text of the manuscript. If you use abbreviations only ones, than there is no need for them. Abbreviations should be defined first time they appeared. Also they should be written in table and figure captions are these parts of the manuscript should be fully understandable without reading the text. Please check the whole paper.
Full names of all the goods, providers and manufactures (of instruments, softwares, reagents and chemicals) should be included
Section 2.4- please check the wavelength?
Section 2.6- units for IC50 are missing in the method description.
According to my opinion table from the Supplement material should be part of the paper, and maybe chromatograms could be supplement. I think that it gets better insight into chemical composition of the samples. Also, the chemical compounds detected could be discussed better and compared with data of previous findings.
I know that this is optional, but according to my opinion the combined sections of “Results and Discussion“ should significantly improve the paper.
Conclusion: Some points of future research can be added.
All references are not in Journal format.
Author Response
26th September 2022
Manuscript ID: foods-1899094
Response to the editors and reviewers of the Foods Journal
Dear Milena Petrovic Markovic,
Regarding reviewers’ comments and suggestions on manuscript entitled “A Thai Traditional triple fruit formulation “Phikud Tri-Phon” may provide fat loss and nutritional benefits” for publication in the Journal of Foods, we do appreciate and grateful for you and reviewer’s valuable statements to improve our manuscript. We have corrected according to all comments by using the track changes in the manuscript. We also wrote the responses for all comments point by points below.
Response to Reviewer 2 Comments
The paper refers to the scientific field covered by the journal.
There are some typos and English needs revision in places.
Thank you for taking the time to read our MS and for your comments
Comment 1: Abstract: line 24: aimed to investigate, Line 25: compositions or better components; Line 28: extracts (plural), ranged instead of were; Line 29 Trolox equivalents (TE), please define also the above used abbreviations (QE, GAE), etc.
Response 1: We have revised in the abstract, line 24, 25, 30-31.
Comment 2: The Abbreviations could be deleted from the abstract as it can stand independently. Also when the phrase is first time used it should be defined in the main text of the manuscript. If you use abbreviations only ones, than there is no need for them. Abbreviations should be defined first time they appeared. Also they should be written in table and figure captions are these parts of the manuscript should be fully understandable without reading the text. Please check the whole paper.
Response 2: We have deleted the abbreviations in the abstract and revised in the whole paper.
Comment 3: Full names of all the goods, providers and manufactures (of instruments, softwares, reagents and chemicals) should be included.
Response 3: We have added in the whole paper.
Comment 4: Section 2.4- please check the wavelength?
Response 4: We have used and revised to the wavelength at 765 nm, page 3, line 134.
Comment 5: Section 2.6- units for IC50 are missing in the method description.
Response 5: We have added in Section 2.6 as µg/mL, page 4, line 159.
Comment 6: According to my opinion table from the Supplement material should be part of the paper, and maybe chromatograms could be supplement. I think that it gets better insight into chemical composition of the samples. Also, the chemical compounds detected could be discussed better and compared with data of previous findings.
Response 6: We have added the table from supplement material into the paper.
Comment 7: I know that this is optional, but according to my opinion the combined sections of “Results and Discussion“ should significantly improve the paper.
Response 7: We have revised from you by combined sections of “Results and Discussion”.
Comment 8: Conclusion: Some points of future research can be added.
Response 8: In old version, we have written in the discussion, Further research is recommended to determine their efficacy and safety in animal and clinical models. However, we have added in the conclusion too.
Comment 9: All references are not in Journal format.
Response 9: We have appropriately reformatted all references.
We had responded to all comments by the reviewers and made changes to the manuscript when it was needed. We hope that the paper is now acceptable for publishing in Foods.
Thank you very much
Yours Sincerely,
Sakan Warinhomhoun, Ph.D. (Corresponding author)
School of Medicine, Walailak University,
222 Thaiburi, Thasala district, Nakhon Si Thammarat 80160, Thailand
Tel.: +66 75677417 E-mail: Sakan.wa@wu.ac.th, sakan.cu@gmail.com

Round 2
Reviewer 1 Report
The authors have addressed the points made.
Reviewer 2 Report
As the majority of reviewer's suggestions have been accepted, I found this article acceptable for publication in its present form.